# Can *Beauveria bassiana* (Bals.-Criv.) Vuill. Control the Key Fruit Pests of the European Chestnut Tree, under Field Conditions?

**DOI:** 10.3390/insects14040342

**Published:** 2023-03-31

**Authors:** Maria Eliza Cota e Souza, Filomena Nóbrega, Albino António Bento

**Affiliations:** 1Mountain Research Center (CIMO), Polytechnic Institute of Bragança, 5300-253 Bragança, Portugal; 2Associate Laboratory for Sustainability and Technology in Mountain Regions, Polytechnic Institute of Bragança, 5300-253 Bragança, Portugal; 3National Institute of Agricultural and Veterinary Research, 2784-505 Oeiras, Portugal

**Keywords:** *Castanea sativa*, carpophagous pests, *Cydia splendana*, *Curculio elephas*, damage, *Beauveria bassiana*, pest management, biological control, field conditions

## Abstract

**Simple Summary:**

Portugal has the largest chestnut tree plantation area for fruit production in Europe. The carpophagous pests *Cydia splendana* Hübner and *Curculio elephas* Gyllenhal are among the main pests that affect nut productivity and quality in the country. Their control is considered difficult due to the large size of the trees, and chestnut orchards are mostly located in mountain regions with high slopes. On the other hand, the biology of the insects, which spend most of their life cycle inside the fruit or buried in the ground, is an aspect that makes them difficult to control. Entomopathogenic fungi are considered safe biological organisms for the environment and capable of regulating the natural population of pests. *Beauveria bassiana* (Bals.-Criv.) Vuill. is one of the most used for this purpose; however, its effectiveness is highly influenced by environmental factors. The study aimed to investigate the efficiency of *B. bassiana* in controlling these two carpophagous pests under environmental conditions. The fungus could infect and kill *C. splendana* and *C. elephas* larvae under the two conidial concentrations tested, showing favorable results for its use in the biocontrol of these insects.

**Abstract:**

The chestnut moth (*Cydia splendana* Hübner) and the chestnut weevil (*Curculio elephas* Gyllenhal) cause serious damage to European producers and companies that transform and market the chestnut. The objective of the present work was to evaluate, under field conditions, the possibilities of *Beauveria bassiana* (Bals.-Criv.) Vuill. to infect and kill the larvae of the two main carpophagous pests of European chestnut, in treatments directed at the soil. For this purpose, the surfaces of vases were sprayed with two concentrations of conidia/mL 5 × 10^7^ (T1) and 1 × 10^8^ (T2). The control (T0) was sprayed with distilled water. Larval mortality and infection were evaluated on five dates (D8 to D220). Confirmation of the fungus present in the larva was performed by molecular analysis. The results obtained are promising for the use of *B. bassiana* as a biological control agent against these key pests of the chestnut crop. There were no significant differences in mortality between the T1 and T2 modalities, however, they were significantly higher than the control. In the case of total mortality (dead and infected larvae), no significant differences were observed for *C. elephas* either. In the case of *C. splendana*, the T2 modality obtained better results in terms of total mortality.

## 1. Introduction

In Europe, the European chestnut tree (*Castanea sativa* Mill.) stands out, in increasing order, in France, Greece, Spain, Italy and Portugal [1], countries that represent more than 90% of the area occupied by chestnut trees for fruit production.

In Portugal, this cultivation area has been increasing in recent decades, from 29,011 ha in 1999 to 50.370 ha in 2021, and Portugal is currently the European country with the largest area of chestnut trees for fruit production. The mountainous areas of the interior north (Trás-os-Montes) concentrate around 85% of the chestnut production area in the country [2]. A significant part of the chestnut tree area is located in mountain regions, cultivated in a multifunctional system for different uses, namely fruit production, timber, mushrooms, recreation and agrotourism, which ensure landscape value and biodiversity. In mountain areas, the chestnut tree is considered the main axis of development based on the tradition of this crop, the favorable environmental conditions and the quality of the varieties.

However, the chestnut tree is a highly vulnerable species and is even considered a threatened species in terms of survival due to biotic and abiotic threats, such as difficulties in adapting to climate change. Introduced through the global trade in timber and nontimber products, or by natural dispersal, a range of invasive pests and diseases have challenged *C. sativa* over the past two centuries, including ink disease (*Phytophthora* spp.), canker (*Cryphonectria parasitica* (Murr. Barr.)) and, more recently, the Asian chestnut gall wasp (*Dryocosmus kuryphilus* (Yasumatsu)) [3,4].

In addition to these threats to the survival of the chestnut tree, carpophagous pests affect the productivity and quality of the chestnut crop. In the Trás-os-Montes region, the chestnut moth caused by the pest complex *Cydia splendana* (Hübner), *Cydia fagiglandana* (Zeller), *Pammene fasciana* L., and the chestnut weevil, *Curculio* (= *Balaninus*) *elephas* (Gyllenhal) in up to 70% of the fruits attacked are identified as the main pests of chestnut groves, causing great damage to the fruit, depreciating their commercial value and causing losses in income for farmers and commercialization and processing companies [5,6,7].

The chestnut moth *C. splendana* is a lepidopteran from the Tortricidae family, with an annual generation, which attacks plants of the genera *Castanea*, *Quercus*, *Fagus* and *Juglans* [8]. Adults emerge in early July, flying until early October. The females lay their eggs on chestnut leaves, located close to chestnut hedgehogs. After the caterpillars hatch, they head towards the chestnut hedgehogs, opening holes in the chestnuts as they feed on the fruit. At the end of development, the caterpillars leave the chestnut and bury themselves in the soil, where they remain until June of the following year, in caterpillar form and later as a pupa.

The weevil, *C. elephas*, is a coleopteran of the Curculionidae family univoltine and can present a diapause that can last up to four years [8]. Its hosts are plants belonging to the genera *Castanea*, *Quercus* and, more rarely, *Corylus*. Adults emerge in August and September. The females introduce the rostrum into chestnut hedgehogs, piercing them with the aid of the mandibles until they find a fruit, where they introduce an egg with the ovipositor (“oviscapto”) [9]. The larva develops inside the chestnut for approximately 40 days. After development, the larva makes a hole in the fruit through which it falls to the ground. Subsequently, it hibernates as the last-instar larva in the soil [9].

The chemical control of these insects is considered complicated due to the size of the chestnut tree, the geography of the groves, generally in areas with high slopes or areas of environmental protection, and the long latent period of the insects. This makes difficult the opportunity for treatment [8], which must be carried out before the larvae penetrate the chestnut hedgehog, which happens between the end of August and the end of September in the region of Trás-os-Montes [7,10]. In addition, the use of plant protection products can lead to the development of resistance in the insect and the contamination of wild mushrooms that are of great importance in association with the activity [11].

Entomopathogenic fungi regulate the natural population of pests, such as insects, ticks and mites [12], acting normally, by contact. *Beauveria bassiana* (Bals.-Criv.) Vuill. is one of the most effective agents in biological control; it is found in all types of soil [13,14]. Different isolates have been identified to attack a wide range of insects (707 species belonging to 15 orders) and mites (13 species) [14,15,16]. According to [17], almost all major taxa of intensively harvested insects are likely considered natural hosts for *B. bassiana* in temperate regions.

Like other entomopathogenic fungi, the action of *B. bassiana* is highly influenced by environmental factors: temperature, humidity and radiation. In addition, host availability and cultural management also affect its growth and its ability to infect. The optimum growth temperature lies between 23–28 °C, with a minimum of 5–10 °C and a maximum of 30–38 °C [16]. For [18], the optimal temperature for the conidial germination of *B. bassiana* isolates was approximately 25 °C, with an upper limit of 30 °C.

Solar radiation is an important factor limiting the effectiveness of entomopathogenic fungi used as biocontrol agents in field applications. According to [19], UV-B radiation caused a significant drop in conidial survival in laboratory and field tests; this same study indicates that it would be necessary to include UV protection additives in the formulation to increase the persistence of *B. bassiana* under exposure to radiation.

Although some disadvantages can be indicated for fungus-based bio-insecticides (speed of death and cost), compared to conventional insecticides [20], they also have considerable advantages. There are no reports of insect pests acquiring resistance to infection by entomopathogenic fungi [21]; fungal isolates from a given insect species tend to be more virulent to that species and may have a narrower host range [22].

Other benefits include safety for mammals, high persistence in the environment, as well as the possibility of modifying, through biotechnological research, the production of enzymes and toxins responsible for infection and mortality [23].

Laboratory studies show the effectiveness of *B. bassiana* on *C. splendana* and *C. elephas* larvae [24,25,26,27,28,29]. Studies evaluating the action of *B. bassiana* under field conditions are scarce and, in the case of the chestnut moth, non-existent. As mentioned, climatic conditions, namely temperature, humidity, and radiation, exercise a strong influence on the physiology of the fungus. Its capacity to infect the host, the progression of the infection inside the living or dead host, the sporulation of the cadaver, the capacity for dissemination and survival of infectious conidial, and also the susceptibility or resistance of the host to infection [30]. In this context, the objective of the present work was to evaluate, under field conditions, the possibilities of *B. bassiana* (Bals.-Criv.) Vuill., strain ATCC74040, to infect and kill the larvae of European chestnut’s two main carpophagous pests, in treatments directed at the ground.

## 2. Materials and Methods

### 2.1. Treatments

The evaluation of the effectiveness of the *B. bassiana* fungus strain ATCC74040 Naturalis^®^ (CBC (Europe) S.r.l. – BIOGARD Division, Cesena, Italy) as a biological control agent against the chestnut moth (*C. splendana*) and the chestnut weevil (*C. elephas*), was carried out in a semifield test (pots) with two concentrations of conidia, 5 × 10^7^ (T1) and 1 × 10^8^ (T2) conidia/mL, and a control (T0). 

With a pipette, the amount of Naturalis^®^ necessary for the intended concentration of spores was removed, after homogenizing the bioinsecticide, which was placed in one liter of distilled water. The application was carried out with a BFL0860 BricoFerr^®^ sprayer. In the control treatment, distilled water was sprayed.

The field portion of the test was carried out between November 2020, when the larvae of both species leave the chestnuts to pupate, and June 2021. The pots were filled with the substrate (soil and perlite at a 1:1 ratio). This mixture was sterilized by autoclaving it at 120 °C for 20 min.

For each modality, 15 repetitions (pots) were used, each with 10 larvae of *C. splendana* and 10 larvae of *C. elephas* (150 larvae of each insect per modality). The vases of each modality were sprayed with the respective concentrations of spores or distilled water, in the case of the control modality.

### 2.2. Insects

The larvae of *C. splendana* (larval stages 4 and 5) and *C. elephas* (larval stage 3) were collected after leaving the fruits and placed on the surface of each pot, previously sprayed with the strain ATCC74040 of *B. bassiana* or distilled water (sprayed 24 h before). The pots were covered with a net to prevent the larvae from leaving or the entry of other animals. A few minutes after placing the larvae in the pots, they buried themselves in the substrate. Subsequently, the pots were placed under natural conditions, in a chestnut grove in Bragança, Portugal (6°45′52.97″ N, 41°46′50.14″ W), at an altitude of 797 m. The pots were buried up to the top to simulate natural conditions as much as possible.

### 2.3. Assessment of Mortality and Contamination of Larvae

The effectiveness of the application of *B. bassiana* on *C. splendana* and *C. elephas* larvae was evaluated on five different dates, at day 08 (D8), day 15 (D15), day 62 (D62), day 100 (D100) and day 220 (D220) after treatment. On each date, 3 pots (replications) of each modality were removed from the field, that is, a total of 30 larvae of each insect/modality were removed.

### 2.4. Fungal Isolate

The larvae of each species, after being collected from the pots (dead or alive), were disinfected and placed in a potato dextrose agar (PDA) culture medium, with the addition of antibiotics (60 mg of streptomycin and 60 mg of penicillin G potassium). The dishes were incubated at 25 ± 1 °C in the absence of light to determine the presence of external sporulation according to [31,32]. After 8 days of incubation, the growth of *B. bassiana* in the larvae was recorded by observing the growth of the fungus on the surface of the larvae in the culture medium and counting the number of contaminated larvae.

The fungus obtained from each larva was transferred to a new PDA dish and placed again at 25 ± 1 °C in the absence of light. This procedure was repeated until pure cultures were obtained, and spore production was observed. Pure isolates were transferred to 50 mL Falcon tubes with PDA tilted to maintain a working collection.

### 2.5. Weather Data

Weather data were collected daily during the period in which the pots were buried in the field by the automatic station (Vantage Pro2 Sensor Suite or (ISS) Weather Station) of the Escola Agrária de Bragança of the Polytechnic Institute of Bragança (ESA/IPB), located at Quinta Santa Apolónia in Bragança, Portugal. The average soil temperature was measured in degrees centigrade (°C) at the surface (TmnG1) and a depth of 15 cm (Tmn15). Precipitation was measured in millimeters (mm/day).

### 2.6. DNA Extraction, PCR Amplification and Sequencing

Confirmation of the fungus strain was performed by molecular analysis. Genomic DNA was extracted from mycelia collected from the agar surface using a DNeasy Plant Mini kit (Qiagen GmbH, Hilden, Germany) according to the manufacturer’s instructions.

Polymerase chain reaction (PCR) was performed to amplify a distinct DNA region, the internal transcribed spacer (ITS) of ribosomal DNA, using a universal primer set for fungi: the forward primer ITS5 and the reverse primer ITS4 [33].

PCR was performed in a 25 µL reaction mixture containing 12.5 μL of Supreme NZYTaq II Master Mix (NZYTECH, Portugal), 1 μL of each primer (10 uM), and 1 μL of template DNA. PCR reactions were performed in a Biometra TGradient thermo cycler (Biometra, Göttingen, Germany) with the following thermal cycling conditions: initial denaturation at 94 °C for 3 min, followed by 30 cycles consisting of denaturation at 94 °C for 30 s, annealing at 55 °C for 30 s and extension at 72 °C for 1 min, and a final extension at 72 °C for 10 min. The PCR products were cleaned using EXO-SAP (Exonuclease I and FastAPTM Thermosensitive Alkaline Phosphatase, Thermo Fisher Scientific Baltics, Vilnius, Lithuania) following the manufacturer’s recommendations. The amplified PCR products were sequenced at Stabvida (Costa da Caparica, Portugal) using the same primers used for their amplification.

Sequences were used as the query at NCBI GenBank for species identification using the BLAST tool (https://blast.ncbi.nlm.nih.gov/, accessed on 7 December 2021).

### 2.7. Data Analysis

The analyses were performed using the RStudio program, version 1.1.17. The results are presented as the original percentage and are the average of the three replicates on each analyzed date. Insect mortality was corrected for the control treatment (natural mortality) using the Schneider-Orelli formula [34] described below:(1)$$\%Mcor=\frac{\left(\%Mobs-\%Mcont\right)\times 100}{100-\%Mcont}$$
where:%Mcor = % of corrected mortality;%Mobs = % of mortality observed in treatment;%Mcont = % of mortality in the control group.

Climate analyses were tabulated in Excel spreadsheets.

Probit regression analysis [35] was used to obtain mean lethal mortality values (LT50) and the respective 95% confidence intervals (95%CI). The program MedCalc Version 20.218 (MedCalc Software, Mariakerke, Belgium) was used to carry out the Probit analyses and build the dose–response curves.

## 3. Results

### 3.1. Weather Data

During the field test period, November 2020 to June 2021, the minimum temperature recorded was in January, which was −10.6 °C at the soil surface and −10.3 °C at 15 cm depth (Figure 1A).

There were no large variations between soil temperature in TmnG1 and Tmn15 (Figure A1), and the average deviation was 0.82 ± 0.06 °C. The temperature remained above 5°C for 64 days in TmnG1 and 71 days in Tmn15, reaching 10°C just 169 days after burying the pots, and remaining above this level for 12 days in TmnG1 and 14 days in Tmn15.

The months with the highest volume of precipitation (Figure A2) were February (183 mm), November (102.6 mm), January (87.8 mm) and April (70 mm).

### 3.2. Efficacy of B. bassiana Strain ATCC74040 (Mortality)

Mortality refers to the percentage of larval mortality on the date of the efficacy evaluation (pot collection date). The concentrations of 5 × 10^7^ (T1) and 1 × 10^8^ (T2) of *B. bassiana* spores showed higher larval mortality for both *C. splendana* (*p* = 0.012) and *C. elephas* (*p* < 0.001) compared to the control modality (T0) (α = 0.05). However, some variation in the data over time was observed. The evaluation of the effectiveness of *B. bassiana* strain ATCC74040 on larvae at D8, D15 and D62 showed mortality rates lower than 25.46 ± 1.19 and 33.33 ± 0.27 for *C. splendana* and *C. elephas*, respectively. At day 220, the percentage of dead larvae was 58.33 ± 1.44% for *C. splendana* and 88.75% for *C. elephas* (Table 1).

### 3.3. Efficacy of B. bassiana Strain ATCC74040 (Total Mortality)

Total mortality refers to dead larvae on the date of the evaluation of effectiveness (the date of the collection of pots), plus infected larvae and pupae (larvae that developed *B. bassiana* in the laboratory), since the temperature of the soil between November and early May was almost always below the minimum for the development of the fungus, and for the fungus to cause the death of insects (Table 2).

The T2 modality showed the highest total mortality of *C. splendana* larvae with the entomopathogenic fungi, followed by T1 and the control treatment, which had the lowest total mortality (*p* < 0.001; α = 0.05). Eight days after treatment, the mortality observed in *C. splendana* larvae was null, both in the T1 and T2 modalities. However, the larvae of *C. splendana* were contaminated, having developed *B. bassiana* by 21.74 ± 0.47% for each treatment modality. In treatment T1, the percentage of total mortality of *C. splendana* larvae ranged from 16.08 ± 0.47% on D62 to 54.55 ± 0.47% on D220. The T2 modality presented total mortality values between 21.74 ± 0.94% (D8) and 88.64 ± 0.47% (D220). The control modality presented total mortality rates between 4.17 ± 0.27% (D8) and 10.71 ± 0.47% (D100).

### 3.4. Mean Lethal Time LT50

Probit regression analyses (dose–response analysis) show that the mean lethal time of *C. splendana* larvae varied little with the concentration, within the confidence interval (Figure 1). In the T1 modality, the LT50 was 157 days (significance level: *p* < 0.0001; chi square: 26.09; DF: 1) while the T2 modality presented an LT50 of 156 days (significance level: *p* < 0.0001; chi square: 26.79; DF: 1).

The mean lethal time of *C. elephas* larvae also varied little with increasing concentrations, within the confidence interval (Figure 2). In the T1 modality, 50% of mortality was observed in 77 days (*p* < 0.0001; chi squared = 42.569; DF = 1)k, and in the T2 modality, the LT50 was reached in 93 days (*p* < 0.0001; chi squared = 37.622; DF = 1).

*Curculio elephas* presented an LT50 that was lower than that of *C. splendana* in the two concentrations tested; in the T1 modality, the larvae of the former died 80 days before those of the latter did, and in the T2 treatment, this occurred in 63 days.

The regression data confirm the ANOVA data that show that there is no significant difference in the mortality of *C. splendana* and *C. elephas* larvae between the two spore concentrations of *B. bassiana* strain ATCC74040 tested and suggest that *C. elephas* is more susceptible.

## 4. Discussion

Most of the studies carried out on the effectiveness of *B. bassiana* against insects of the Curculionidae (Coleoptera) and Tortricidae (Lepidoptera) families have been conducted under laboratory conditions (e.g., [26,27,29,36,37], and our study is a pioneer in testing the possibilities of *B. bassiana* in controlling *C. splendana* and *C. elephas*, in groves under real environmental conditions on the Iberian Peninsula.

Several laboratory studies show the effectiveness of *B. bassiana* on insects of the Tortricidae and Curculionidae families tested in a controlled environment. The authors of [37] reported that *B. bassiana* may be a potential biological control agent against *Grapholita molesta* (Busck). The authors of [36] observed mortality rates between 57.6% and 84.6% for three isolates of *B. bassiana* in *Thaumatotibia leucotreta* (Meyrick). The authors of [29] obtained a larval mortality of *C. elephas* with the commercial strain ATCC 74040 of 77% after 14 days and that of 80% after 28 days. The authors [26] tested four isolates and different concentrations of *B. bassiana* in *C. splendana* larvae in the laboratory and obtained the best results with the concentrations 5 × 10^7^ and 1 × 10^8^ conidia/mL.

Some field tests corroborate the findings obtained in the laboratory. In [38], a field study on peanut pests in India concluded that the use of *B. bassiana* increased production and showed persistence in the field for the next crop. In [39], a field trial carried out in Italy between October and May 2002 concluded that soil treated with *B. bassiana* showed a 35% increase in the mortality of *C. elephas* larvae.

Generally, entomopathogenic fungi survive well in nature between 10 °C and 30 °C [40]. *Beauveria bassiana* has an optimum growth temperature of around 23–28 °C, a minimum growth temperature of 5–10 °C and a maximum growth temperature of 30–38 °C [16]. Temperature-dependent phenomena such as the rate of disease development and disease level are greater at optimum temperature [40].

Soil temperature conditions may represent one of the main reasons for the high LT50 of the two species when compared to those of laboratory tests, where *B. bassiana* remains in optimal conditions of temperature and humidity. Only 200 days after the treatments were applied, the soil temperature remained above 10 °C (Figure 1A). However, the important thing to note when using *B. bassiana* as a microbiological control method for these two species is that when the larvae penetrate the soil, they are contaminated. Later, when the soil reaches the temperature and humidity suitable for development, the fungus will grow and cause the insects to die before their emergence, which in the Trás-os-Montes region occurs at the end of July.

The ability of *B. bassiana* strain ATCC74040 to infect *C. splendana* and *C. elephas* larvae at the time they bury themselves in the soil to pupate was evident in the results of the total mortality, as well as data from the last sampling (D220) on 06/15, with mortality rates of 58.33% (T1) and 88.75% (T1), respectively, for *C. splendana* and *C. elephas*. The results also point to the presence of *B. bassiana* in 88% (T2) of *C. splendana* larvae and 88.75% (T1 and T2) of *C. elephas*.

Given the results obtained, we consider that the control of *C. splendana* and *C. elephas*, with *B. bassiana* strain ATCC74040, will have greater chances of success if directed at the soil, before the larvae leave to pupate (October), than spraying the chestnut tree does, at the time when the larvae hatch and penetrate the hedgehogs (end of August and September), which coincides with a period of very low atmospheric humidity and very high solar radiation, the main factors of inactivation of *B. bassiana*. There is also the difficulty of treating large trees and groves installed in areas with large slopes. Application to the soil, at the time the chestnuts fall (end of October), with spraying only in the projection of the tree canopy, allows for easier, faster work and better conditions for the fungus to survive for longer periods. At that time, solar radiation is much lower, and soil moisture is high and allows contact between the fungus and the larvae when they leave the chestnut and bury themselves in the soil.

This treatment coincides with the less lethal environmental conditions for the fungus with its low temperatures, which slow down the development of the fungus inside the insects and their death. The effectiveness of the entomopathogenic fungus under these study conditions must be considered during the period that the insects spend buried in the soil; it greatly exceeds the values obtained in laboratory tests; however, there is a low emergence of adults in the following spring.

## 5. Conclusions

The test carried out in pots with autoclaved soil confirmed that the fungal strain ATCC 74040 of *B. bassiana* was capable of infecting and killing larvae of *C. splendana* and *C. elephas* at both concentrations of conidia, in semifield conditions.

The present study presents promising performance results for the use of *B. bassiana* strain ATCC 74040 as a biological control agent against carpophagous insects, key pests of the chestnut crop, with total mortality rates of *C. splendana* and *C. elephas* larvae being close to 90%.

## Figures and Tables

**Figure 1 insects-14-00342-f001:**
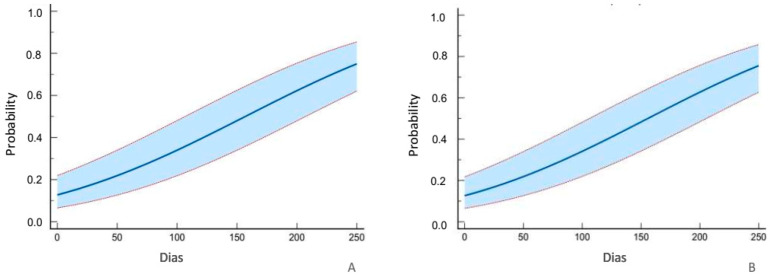
Probit regression of chestnut moth, *Cydia splendana* (Hübner), in T1 (**A**) and T2 (**B**) modalities.

**Figure 2 insects-14-00342-f002:**
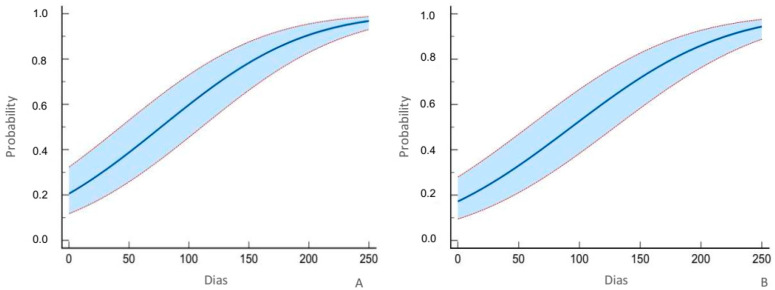
Probit regression of chestnut weevil, *curculio elephas* (Gyllenhal) in T1 (**A**) and T2 (**B**) modality.

**Table 1 insects-14-00342-t001:** Mortality percentage of chestnut moth, *Cydia splendana* (Hübner), (A) and chestnut weevil, *Curculio elephas* (Gyllenhal) (B) larvae in different modalities on days 08, 15, 62, 100 and 220 after treatment. Insect mortality was corrected for the control treatment (natural mortality) using the Schneider-Orelli formula.

	*Cydia splendana* (A)						***p*-value**
		**D8 c**	**D15 bc**	**D62 bc**	**D100 ab**	**D220 a**	<0.001
	**T0 B**	0	0 A	4.17 ± 0.54 A	10.71 ± 0.47 A	12 ± 0 B	0.3639
	**T1 A**	0 c	25 ± 1.25 Abc	19.73 ± 0.47 Abc	38 ± 1.41 Aab	54.55 ± 0.47 Aa	0.0178
	**T2 A**	0 c	29.17 ± 0.7 Ab	25.46 ± 1.19 Ab	36.53 ± 1.44 Ab	58.33 ± 1.44 Aa	0.05
***p*-value**	0.012	-	0.6249	0.3984	0.1991	0.0216	
	*Curculio elephas* (B)						***p*-value**
		**D8 b**	**D15 b**	**D62 b**	**D100 a**	**D220 a**	**<0.001**
	**T0 B**	0 A	12.5 ± 0.27 A	12.5 ± 0.47 B	11.11 ± 0.47 B	11.11 ± 0.47 B	0.5251
	**T1 A**	6.67 ± 0.27 Ad	7.64 ± 0.72 Ac	22.29 ± 0.72 Ab	83.93 ± 0 Aa	85 ± 0.27 Aa	<0.001
	**T2 A**	3.33 ± 0.27 Ad	26.11 ± 0.82 Ac	33.33 ± 0.27 Ac	61.21 ± 0.72 Ab	88.75 ± 0 Aa	<0.001
***p*-value**	<0.001	0.2963	0.182	0.0493	<0.001	<0.001	

A and B: different capital letters in columns indicate statistical difference by *t*-test (*p* < 0.05); a and b: different lowercase letters in the lines indicate statistical difference by the *t*-test (*p* < 0.05); equal letters do not differ significantly from each other.

**Table 2 insects-14-00342-t002:** Percentage of total mortality (including death and infection) of chestnut moth, *Cydia splendana* (Hübner) (A), and chestnut weevil, *Curculio elephas* (Gyllenhal) (B) larvae in the different modalities on days 08, 15, 62, 100 and 220 after treatment. Insect mortality was corrected for the control treatment (natural mortality) using the Schneider-Orelli formula.

	*Cydia splendana* (A)						***p*-value**
		**D8 b**	**D15 b**	**D62 b**	**D100 b**	**D220 a**	0.0064
	**T0 C**	4.17 ± 0.27 A	6.67 ± 0.27 B	8.33 ± 0.27 B	10.71 ± 0.47 B	10.71 ± 0 C	0.5339
	**T1 B**	21.74 ± 0.47 Ab	25.11 ± 0.27 Ab	26.63 ± 0.47 Bb	46.53 ± 0.72 Ab	54.55 ± 0.47 Ba	0.0017
	**T2 A**	21.74 ± 0.94 Ab	37.5 ± 0.27 Ab	45.45 ± 0.72 Ab	40 ± 0.47 Ab	88.64 ± 0.47 Aa	0.0014
***p*-value**	<0.001	0.2843	0.0091	0.0085	0.039	<0.001	
	*Curculio elephas* (B)						***p*-value**
		**D8 b**	**D15 b**	**D62 b**	**D100 a**	**D220 a**	**<0.001**
	**T0 B**	0 B	3.33 ± 0.27 A	8.33 ± 0.27 A	7.4 ± 0 B	11.11 ± 0.47 B	0.5251
	**T1 A**	10 ± 0.82 Ab	17.95 ± 0.82 Ab	25.82 ± 0.54 Ab	88.42 ± 0.27 Aa	88.75 ± 0 Aa	<0.001
	**T2 A**	3.33 ± 0.27 Bc	28.66 ± 0.47 Ab	36.36 ± 0.98 Ab	73.93 ± 0.27 Aa	88.75 ± 0.72 Aa	<0.001
***p*-value**	<0.001	0.0247	0.087	0.1322	<0.001	<0.001	

A and B: different capital letters in columns indicate statistical difference by *t*-test (*p* < 0.05); a and b: different lowercase letters in the lines indicate statistical difference by *t*-test (*p* < 0.05); equal letters do not differ significantly from each other.

## Data Availability

The data presented in this study are available upon request from the corresponding author. The data are not publicly available due to their relevance to an ongoing Ph.D. thesis.

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
