# Peer review of "Can *Beauveria bassiana* (Bals.-Criv.) Vuill. Control the Key Fruit Pests of the European Chestnut Tree, under Field Conditions?"

_insects, 2023, doi:10.3390/insects14040342_

Round 1

Reviewer 1 Report

Manuscript ID 2286720: Can Beauveria bassiana (Bals.-Criv.) Vuill control the key fruit pests of the European chestnut tree, under field conditions? This manuscript reports an interesting study, with relevant contribution to the biological control of pests. However, the text needs a deep revision before the study be considered for publication. Some of my concerns are listed below.  

Major comments:

Many details are missing in the Material and Methods section:

1) the authors must report how fungal conidia were produced and formulated;

2) the authors must provide the method and the volume of conidial suspension used in each treatment; I mean, the equipment and the quantity of conidia sprayed per area, per soil surface;

3) Was the relative humidity measured during the field test? Also, describe if the area was protected from the solar irradiation; this is also an important information;

4) What about the viability of conidia? Was it checked before application? Was the viability of conidia checked after application by isolation of the fungus from the soil?;

5) Fungal isolation from dead insects is usually done by placing the dead insect in a moist chamber before transferring the fungi to a culture medium plate. Why did the authors opt for placing the larvae straight to the culture media? Is there a reason for this? In fact, viable conidia on the larva surface, even those that did not cause infection (e.g., maybe even those that were applied to the soil), were able to grow in the PDA medium.

6) The confirmation of the fungal strain was conducted with how many isolates obtained from the infected larvae? All? Are the primers tested specific to the ATCC 74040? Because if the purpose of the test is the confirmation of the cause of death (re-isolation of the fungal strain sprayed), the primers have to differentiate the applied strain from other indigenous Beauveria in the soil. Right?

Results:

1) Why in table 1, some means are not followed by a capital letter. Is there a mistake here? The table present corrected mortality? Please indicate that in the title.

2) Lines 236-237: Did the authors check if the infection was a contamination with ATCC 74040, or infection by indigenous B. bassiana? This fungus is a common inhabitant from the soil, as the authors mentioned in Introduction

3) Ideal temperature for B. bassiana development and infection was not reported on the first days after soil treatment. How the authors explain the larval mortality in the first days after treatment (lines 140-141)?

Discussion:

In my opinion Discussion should not be divided in subsections. Please, revise it. Also, the authors limited most of the Discussion section by presenting and comparing the percentage of mortality of previous studies with the results obtained in the current study, which I assume make it less strong.

Conclusions:

The conclusions section is too long. In fact, most of the text would fit better in the Discussion section.

Minor comments:

Line 20, 87 and elsewhere: Correct “conidial concentrations”;

Lines 68-73: Long period. Revise text.

Lines 80, 87, 283, 285, and elsewhere: Revise the correct form for making citation in the text.

Line 103: Reference 28 is duplicated.

Line 106: Revise language.

Line 110: The genus Beauveria must be abbreviated.

Lines 278-279: Revise text.

line 289: This sentence has no meaning.

Author Response

The authors express their gratitude to the reviewer for providing valuable comments and suggestions that helped enhance the quality of the manuscript.

Reviewer 2 Report

This is a well conceived study presenting results that are clear and well supported but that are also not especially divergent from what might be expected for this fungus and these target hosts. The Materials and Methods seemed to be notably lacking in details about some of the materials and how the studies were done. 

One scientific point that was a bit concerning was why only the one isolate was used for these studies. Yes,  it is a commercialized isolate that is widely available as Naturalis, but there are thousands of other Beauveria bassiana isolates in culture collections around the world, and there is no reason to believe that the isolate studied here is superior to all other possible strains. 

Another issue related to the identity of the fungus used is that the molecular work used to confirm the recovery of Beauveria from the hosts was confined to the use of ONLY the ITS gene; this is worrying because the ITS gene shows very little variability and is notoriously unreliable for identifications of hypocrealean entomopathogens such as B. bassiana. The dependence on only the ITS may have masked the presence of some other indigenous species of Beauveria that might have had activity against the target hosts in the soil. Is it likely that this 'problem' puts the overall conclusions of this study in doubt? Most probably no, but this is a question that will remain unanswered.

There are a considerable number of editorial changes that have been suggested on the attached PDF file of the submitted manuscript. All of the changes suggested are in the comments appended to the file. Several of these comments note issues with the content of the text rather than just the mechanics of its presentation.

Author Response

(The authors gave the same response as above.)

Reviewer 3 Report

The draft of article number 2286720 submitted to the Insects, MDPI, entitled “Can Beauveria bassiana (Bals.-Criv.) Vuill control the key fruit pests of the European chestnut tree, under field conditions” carried results in the text that needs some revision for the improvement of the draft. Some suggested changes for example are in the comments portion to revise and improve the manuscript. Please find suggested corrections, reference writing, journal-style format, author’s instructions, use of abbreviations and missing information for revision. 

Can Beauveria bassiana (Bals.-Criv.) Vuill control the key fruit pests of the European chestnut tree, under field conditions? Please revise the title

Line 13-15:  Its control is considered …. the chestnut or buried in the ground” the sentence is lengthy, please rephrase the sentence

Line 58-59: ))” please recheck these

Line 62: Laspeyresia (= Cydia) splendana” please recheck these

Line 130: (L4/L5) and C. elephas (L3)” explain the abbreviations

Line 141-142: (D8), 15 (D15), 62 (D62), 100 (D100) and 220 (D220)” please explain what is this. This is confusing for the readers, please explain the abbreviations where first used

Line 145: PDA” what is this

Line 156: weather data” please correct the words    

Line 166: (ITS)” please explain the abbreviation where first used

Data analysis: how the weather data was analyzed, is missing in the data analysis

Tables 1 and 2: There are two tables in table 1: please explain which is A and B portion” please also explain the abbreviations used

Please use. or,” but you need to be consistent throughout the MS

Please also maintain the spacing between the words” consult the author’s instruction

Line 195-197: Probit regression analysis was used to obtain mean lethal mortality values (LT50)  and respective 95% confidence intervals (95%CI). The MedCalc® program was used to 196 carry out the Probit analyses and build the dose-response curves.” Why the authors calculated the LT50 and why not LC50.

Weather data is also not analyzed

4.17±0.27%” please keep the spacing between words” authors’ instructions

Supplementary figure: please revise the supplementary figure by scaling and or some other way to visualize both lines in the figure. Presently this figure is not readable due to the overlapping of the data

Is there any effect on the plant by these concentrations of B. bassiana used in the present research?

Precise the conclusion as this is lengthy

Supplementary figure: Please also recheck the Dez, fev, mai etc” these abbreviations are not understandable

Line 120-123: The test was carried out between the end of October 2020……substrate (soil and perlite in a 1:1 ratio). This mixture was sterilized by autoclaving at 120 °C for 20 minutes.” October 2020 data on temperature is missing in the figure

 Why the authors didn’t include the humidity and rain data during the period of research

Please double-check for inconsistencies in Journal style/formatting/ authors instructions, double spaces, spellings of the words, English vocabulary, missing italics, scientific names, excessive/missing information, etc.

Author Response

(The authors gave the same response as above.)

Round 2

Reviewer 1 Report

The manuscript was considerably improved. All my queries were satisfactorily addressed. I still listed a few comments below:

Line 30: Correct: “concentrations of conidia/ml”

Line 139-140: Correct: “concentrations of conidia” and “conidia/ml”

Line 328: Revise sentence

Line 341: Correct: “conidia/ml”

Line 392: Correct: “concentrations of conidia”

Author Response

(The authors gave the same response as above.)

Reviewer 3 Report

The draft of article number 2286720 submitted to the Insects, MDPI, entitled “Can Beauveria bassiana (Bals.-Criv.) Vuill control the key fruit pests of the European chestnut tree, under field conditions” carried interesting results in the text and also improved after revision.

Author Response

(The authors gave the same response as above.)
